# Ganoaustralins A and B, Unusual Aromatic Triterpenes from the Mushroom *Ganoderma australe*

**DOI:** 10.3390/ph15121520

**Published:** 2022-12-06

**Authors:** Lin Zhou, He-Ping Chen, Xinyang Li, Ji-Kai Liu

**Affiliations:** 1School of Pharmaceutical Sciences, South-Central Minzu University, Wuhan 430074, China; 2Graduate School of Pharmaceutical Sciences, The University of Tokyo, Bunkyo-ku, Tokyo 113-0033, Japan

**Keywords:** *Ganoderma australe* (Fr.) Pat., Polyporaceae, triterpenes, anti-BACE1 activity

## Abstract

Two triterpenes, ganoaustralins A (**1**) and B (**2**), featuring unprecedented 6/6/6/5/6 scaffolds were isolated from the fruiting bodies of the mushroom *Ganoderma australe*. The structures were determined by extensive NMR and HRESIMS spectroscopic analysis. The absolute configuration of the C-25 in ganoaustralin A was assigned by the phenylglycine methyl ester (PGME) method. The relative and absolute configurations of the polycyclic backbones were determined by NMR and ECD calculations, respectively. The plausible biosynthetic pathways of ganoaustralins A and B were proposed. Ganoaustralin B showed weak inhibition against *β*-secretase 1.

## 1. Introduction

*Ganoderma* is a group of Polyporus fungi with hundreds of species widely distributed in the North Hemisphere [1,2]. There are 460 records of *Ganoderma* on the website Index Fungorum (http://www.indexfungorum.org, accessed on 6 December 2022). Being one of the famous Traditional Chinese Medicines which has been used for centuries, the study of secondary metabolites of this genus has long been a hot topic [3]. In recent years, more and more research has demonstrated that *Ganoderma* is a prolific reservoir for triterpenes [4,5], meroterpenes [6,7,8,9,10,11,12,13,14,15,16], sesquiterpenes [17], alkaloids [18], and steroids [17,19]. Among the reported structures, the triterpenes account for the largest number of chemical entities, and meroterpenes are reported to have diverse scaffolds. Notably, the triterpenes originated from *Ganoderma* occupy half of the proportion of the triterpenoid scaffolds reported from fungi both in amounts and types [20]. The types of triterpenes from *Ganoderma* are mainly lanostanoids and their skeletal variants with many additional modifications, such as new C–C bond formation [21,22], C–C bond cleavage [23], migration [18,24], and degradation [24] (Appendix A). Notably, amongst the post-skeletal formation modifications, the new C–C bond formations are the most important ways for the generation of unprecedented genuine skeletons. So far, only two examples have been reported to have the additional new C–C formation based on the lanostane skeleton, the C-1–C-10 connection in methyl ganosinensate A [22], and C-12–C-23 bond in ganorbifate A [21].

*G. australe*, which mainly dwells in tropic areas, is a central species in the “*G. applanatum-australe* complex” and can be distinguished from *G. applanatum* by possessing larger basidiospores^1^. Compared to other species, the chemical composition of this fungus has been poorly investigated. A few publications on this fungus have shed light on the chemical types of secondary metabolites are also triterpenes and meroterpenes [25,26,27,28]. The structural features of lanostanes from this fungus were more similar to the compounds from *G. applanatum*, while less similar to those from *G. lucidum*. Inspired by the medicinal values and diverse structural scaffolds found in *Ganoderma*, a sample of *G. australe* collected from the rain forest in Yunnan Province (China) has been chemically studied by our group [29]. We herein report the isolation, structural elucidation of ganoaustralins A (**1**) and B (**2**) (Figure 1), and two novel triterpenes with an undescribed benzene ring from *G. australe*.

## 2. Results

### 2.1. Structural Elucidation

Compound **1** was isolated as a pale-yellow gum. The molecular formula of **1** was determined to be C_30_H_34_O_7_ based on the HRMS(ESI) analysis (*m*/*z* [M + Na]^+^ calcd for C_30_H_34_O_7_Na, 529.22022; found 529.21973). The ^1^H NMR spectra of **1** recorded in chloroform-*d* presented partially overlapped signals which were crucial for correct structural elucidation (Table 1). The ^13^C NMR spectra measured in pyridine-*d*_5_ clearly displayed 30 carbon resonances ascribable to five methyl singlets, one methyl doublet, four methylenes, six methines (three olefinic ones), 14 proton-free carbons including five sp^3^ hybridized ones, and three carbonyls (Table 1). The number of carbon resonances and the poly-methyl singlets were reminiscent of triterpenes, the main constituents of the genus *Ganoderma*. Analysis of the ^1^H-^1^H COSY and HMBC spectra indicated that parts of the signals showed resemblance to those of applanoxidic acid C, a lanostane triterpene which was reported from *G. australe* and *G. applanatum* [30]. Specifically, the construction of A–D rings, including the absolute configurations of the chiral centers of **1**, is the same as those of applanoxidic acid C. However, the remaining signals, which consist of two double bonds, a carboxylic group, a doublet methyl group, a methine, and a methylene, are quite different from those of applanoxidic acid C. Further analysis of the 2D spectra allowed the complete structural elucidation of **1**.

The relative configurations of the chiral centers of **1** except for C-25 were determined to be the same as those of applanoxidic acid C [30], including the β orientation of the epoxy ring of C-7 and C-8, which was evidenced by the key ROESY correlations between H-7 (*δ*_H_ 5.07) and H_3_-18 (*δ*_H_ 1.58) (Figure 2). The stereochemistry of C-25 was determined by the phenylglycine methyl ester (PGME) method [31]. The C-26 (*S*)- and (*R*)-PGME derivatives (**1a** and **1b**) of **1** were synthesized by using the corresponding (*R*)- and (*S*)-phenylglycine methyl esters (Figure 3A), ^1^H NMR data analysis of the Δ*δ* (*δ*_S_-*δ*_R_) values of the two synthetic isomers revealed that the C-25 was *S* configuration (Figure 3B). Since the scaffold of **1** was previously undescribed, the structural correctness was further corroborated by the calculation of the ^1^H and ^13^C NMR. As shown in Figure 4A, the regression analysis between the calculated and experimental NMR data gave the *R*^2^ value of 0.9981 for ^13^C NMR data and 0.9943 for ^1^H NMR data (Appendix A), thereby confirming the structural solidity. The absolute configuration of **1** was assigned by ECD calculation. As shown in Figure 4C, the calculated ECD coincides with the experimental CD both in signs and patterns. Therefore, the structure of compound **1** was determined as shown in Figure 3, and it was given the trivial name ganoaustralin A.

Compound **2**, a pale-yellow oil, has a molecular formula of C_31_H_36_O_6_ as indicated by the HRMS(ESI) analysis (*m*/*z* [M + Na]^+^ calcd for C_31_H_36_O_6_Na, 527.24096; found 527.24017). The 1D NMR spectra of **2** presented signals for six methyl singlets (one methoxy group), a methyl doublet, four methylenes, two sp^3^ methines, four sp^3^ quaternary carbons, five double bonds, and four carbonyl groups (Table 1). The above-mentioned data, along with the 2D NMR correlation features, showed great similarity to those of compound **1**, suggesting the analogous structures between **1** and **2**. The structural assignment of **2** was accomplished by interpretation of the 2D NMR spectra. In the HMBC spectrum, key correlations from the olefinic proton at *δ*_H_ 7.56 (H-7) to C-5 (*δ*_C_ 49.7), C-6 (*δ*_C_ 24.1), C-9 (*δ*_C_ 163.2), and C-14 (*δ*_C_ 56.0) (Figure 2) suggested that C-7 and C-8 of **2** was a double bond instead of being an epoxy ring of **1**. In addition, the HMBC correlation from the methyl singlet at *δ*_H_ 3.59 to the carbonyl group at *δ*_C_ 176.7 (Figure 2) revealed that the C-26 carboxylic group was methyl esterified in **2** compared to that of **1**. Therefore, the planar structure of **2** was determined, as shown in Figure 2. The relative stereochemistry of **2** was determined to be the same as that of **1** by analysis of the ROESY spectrum, and by consideration of the biosynthetic pathways and confirmed by ^1^H and ^13^C NMR calculations (Figure 4B). The absolute configuration of **2** was determined by comparison of the calculated ECD and the experimental CD (Figure 4C). Therefore, compound **2** was named ganoaustralin B.

### 2.2. Proposed Biosynthetic Pathway of ***1*** and ***2***

Given that ganoaustralins A (**1**) and B (**2**) represent a new class of triterpene natural products, the plausible biosynthetic pathway was developed as shown in Figure 1. The structure **A**, an analogue of applanoxidic acid C with additional C-21 ester modification [32,33], was proposed to be the precursor. Firstly, the enol form of **A** undergoes an intramolecular Claisen condensation to give the key intermediate **B** with a cyclopropanone moiety. The abstraction of H-17 by base leads to the ring-opening of the cyclopropanone moiety in **B** to obtain **C**. Likewise, the abstraction of H-16 of intermediate C triggers the intramolecular aldol condensation to make the C–C bond between C-16 and C-23 in **D**. Furthermore, the abstraction of H-16 again of **D** produces the intermediate **E** via an E1cb mechanism. Finally, aromatization (enolization) of **E** yields compound **1**, which is further oxygenated and methyl esterified to give compound **2**.

### 2.3. Biological Activity Evaluation of ***1*** and ***2***

The two compounds were screened for biological activity in a panel of bioassays, including cytotoxicity against the cancer cell lines, the inhibition on human protein tyrosine phosphatase 1B (PTP1B), *α*-glucosidase, and *β*-secretase 1 (BACE1) (Appendix A). As a result, only compound **2** showed 44.7% inhibition on BACE1 at the concentration of 40 μM.

## 3. Discussion

Two unprecedented 6/6/6/5/6 polycyclic triterpenes, ganoaustralins A (**1**) and B (**2**), were isolated and identified from the medicinal mushroom *G. australe*. By using NMR elucidation, ^1^H, ^13^C, and ECD calculations, the structures as well as the absolute configurations were unambiguously determined. The chemo-diversity of the triterpenoids is not as varied as the natural sesquiterpenoids and diterpenoids. This can be reasoned by the cyclization of the communal precursor squalene, which is produced by two molecules of farnesyl pyrophosphate by tail–tail connection, is limited by the molecule size and chair-boat conformations. However, the post-oxygenation and resultant carbon degradation, migration, and new bond formation have increased the chemodiversity to a great extent. The report of these two novel triterpenes opens new avenues for the potential of *Ganoderma* in producing structurally intriguing triterpenoids.

## 4. Materials and Methods

### 4.1. General Experimental Procedures

Optical rotations were obtained on an Autopol IV-T digital polarimeter (Rudolph, Hackettstown, NJ, USA). UV spectra were recorded on a Hitachi UH5300 spectrophotometer (Hitachi, Tokyo, Japan). CD spectra were measured on a Chirascan Circular Dichroism Spectrometer (Applied Photophysics Limited, Leatherhead, Surrey, UK). In addition, 1D and 2D spectra were obtained on Bruker Avance III 600 MHz spectrometer (Bruker Corporation, Karlsruhe, Germany). HRESIMS spectra were measured on a Q Exactive Orbitrap mass spectrometer (Thermo Fisher Scientific, Waltham, MA, USA). Medium pressure liquid chromatography (MPLC) was performed on an Interchim system equipping with a column packed with RP-18 gel (40–75 μm, Fuji Silysia Chemical Ltd., Kasugai, Japan). Preparative high performance liquid chromatography (prep-HPLC) was performed on an Agilent 1260 Infinity Ⅱ liquid chromatography system equipped with a Zorbax SB-C18 column (particle size 5 μm, dimensions 150 mm × i.d. 9.4 mm, flow rate 5 mL·min^−1^) and a DAD detector (Agilent Technologies, Santa Clara, CA, US). Sephadex LH-20 (GE Healthcare, Sweden) and silica gel (200–300 mesh, Qingdao Haiyang Chemical Co., Ltd., Qingdao, China) were used for column chromatography (CC).

### 4.2. Fungal Material

The fruiting bodies of *Ganoderma australe* were collected in Tongbiguan Natural Reserve, Dehong, Yunnan Province, China, in 2016, and identified by Yu-Cheng Dai (Institute of Microbiology, Beijing Forestry University, Beijing, China). A voucher specimen of *G. australe* was deposited in the Mushroom Bioactive Natural Products Research Group in South-Central University for Nationalities.

### 4.3. Extraction and Isolation

The dry fruiting bodies of *Ganoderma australe* (3.26 kg) were grounded and extracted four times by CHCl_3_:MeOH (1:1) at room temperature to obtain a crude extract which was further resuspended in distilled water and partitioned against ethyl acetate (EtOAc) to afford EtOAc extract (130 g). The EtOAc extract was eluted on MPLC with a stepwise gradient of MeOH in H_2_O (20%–100%) to afford eight fractions (A−H).

Fraction E was separated by Sephadex LH-20 (CHCl_3_:MeOH = 1:1) to afford four subfractions (E_1_-E_4_). Subfraction E_2_ was separated by column chromatography (CC) on silica gel (petroleum ether–acetone from *v*/*v* 15:1 to 1:1) to obtain 10 subfractions (E_2-1_–E_2-10_). Compound **1** (5.2 mg, *t*_R_ = 14.0 min) was purified from E_2-7_ by prep-HPLC (MeCN-H_2_O: 30:70–50:50, 25 min, 4 mL·min^−1^).

Fraction F was separated by Sephadex LH-20 (MeOH) to afford five subfractions (F_1_-F_5_). Subfraction F_2_ was separated by column chromatography (CC) on silica gel (petroleum ether–acetone from *v*/*v* 15:1 to 1:1) to obtain 13 subfractions (F_2-1_–F_2-13_). Compound **2** (0.8 mg, *t*_R_ = 15.3 min) was purified from F_2-5_ by prep-HPLC (MeCN−H_2_O: 30:70–50:50, 25 min, 4 mL·min^−1^).

*Ganoaustralin A* (**1**): Pale-yellow oil; [α]^25^_D_ +503.6 (*c* 0.35, MeOH); UV (MeOH) λmax (log ε) 205 (4.30), 235 (4.34); ^1^H NMR (600 MHz, C_5_D_5_N) data, see Table 1, ^13^C NMR (150 MHz, C_5_D_5_N) data, see Table 1; HRMS(ESI) *m*/*z* [M + Na]^+^ Calcd for C_30_H_34_O_7_Na 529.22022, found 529.21937.

*Ganoaustralin B* (**2**): Pale-yellow oil; [α]^25^_D_ +234.2 (*c* 0.50, MeOH); UV (MeOH) λmax (log ε) 210 (2.95), 230 (3.97), 285 (4.10); ^1^H NMR (600 MHz, CDCl_3_) data, see Table 1, ^13^C NMR (150 MHz, CDCl_3_) data, see Table 1; HRMS(ESI) *m*/*z* [M + Na]^+^ Calcd for C_31_H_36_O_6_Na 527.24096, found 527.24017.

### 4.4. Biological Activity Assays

Compounds **1** and **2** were subjected to biological assays including cytotoxicity against five human cancer cell lines [34], inhibition on human protein tyrosine phosphatase 1B (PTP1B) [35], *α*-glucosidase [36], and *β*-secretase 1 (BACE1) [37,38]. The cancer cell lines used in this study were the human myeloid leukemia HL-60 (ATCC CCL-240), the human hepatocellular carcinoma SMMC-7721, the human lung cancer A-549 (ATCC CCL-185), the human breast cancer MCF-7 (ATCC HTB-22), and the human colon cancer SW480 (ATCC CCL-228). The SMMC-7721 cell line was bought from China Infrastructure of Cell Line Resources (Beijing, China), and other cell lines were bought from American Type Culture Collection (ATCC, Manassas, VA, USA). The assay procedures are the same as previously reported.

### 4.5. Synthesis of the PGME Derivatives of ***1***

To a DMF (1.0 mL) solution of **1** (0.5 mg, 1.0 μmol), add PyBOP (12.5 mg, 24.0 μmol), HBTU (9.3 mg, 24.5 μmol), DMAP (1.5 mg, 12.3 μmol), and (*S*)-PGME (5.0 mg, 30.3 μmol), and the mixture was stirred at room temperature for 3 h. The solution was diluted with EtOAc (1 mL) and washed with H_2_O. The organic layer was concentrated under reduced pressure to obtain pale-yellow oil, which was purified by HPLC to furnish (*S*)-PGME amide derivative **1a**. Similarly, (*R*)-PGME amide derivative **1b** was prepared from **1** (0.5 mg) and (*R*)-PGME (5.0 mg) in the same conditions. NMR assignments of the protons for (*S*)- and (*R*)-PGME of **1** were achieved by analysis of their ^1^H-^1^H COSY spectra.

**1a**: ^1^H NMR (600 MHz, CDCl_3_), *δ*_H_ 2.176 (1H, overlapped, H-1a), 2.029 (1H, m, H-1b), 2.727 (1H, m H-2a), 2.515 (1H, overlapped, H-2b), 2.851 (1H, dd, *J* = 12.8, 3.0 Hz, H-5), 2.195 (1H, overlapped, H-6a), 1.786 (1H, dd, *J* = 14.8, 12.8 Hz, H-6b), 4.744 (1H, d, *J* = 3.8 Hz, H-7), 6.177 (1H, s, H-11), 1.460 (3H, s, H-18), 1.155 (3H, overlapped, H-19), 7.361 (1H, overlapped, H-20), 6.664 (1H, d, *J* = 2.2 Hz, H-22), 3.185 (1H, dd, *J* = 13.4, 7.5 Hz, H-24a), 3.035 (1H, dd, *J* = 13.4, 6.7 Hz, H-24b), 2.523 (1H, overlapped, H-25), 1.155 (3H, overlapped, H-27), 1.138 (3H, s, H-28), 1.108 (3H, s, H-29), 1.347 (1H, s, H-30), 6.800 (1H, d, *J* = 6.7 Hz, NH), 5.460 (1H, d, *J* = 6.7 Hz, H-2′ of PGME), 7.357 (5H, overlapped, phenyl protons of PGME), 3.700 (3H, s, OCH_3_). HRMS(ESI) *m*/*z* [M + H]^+^ Calcd for C_39_H_44_O_8_N 654.30669, found 654.30615.

**1b**: ^1^H NMR (600 MHz, CDCl_3_), *δ*_H_ 2.176 (1H, overlapped, H-1a), 2.030 (1H, m, H-1b), 2.728 (1H, m H-2a), 2.500 (1H, m, H-2b), 2.854 (1H, dd, *J* = 12.7, 2.9 Hz, H-5), 2.228 (1H, overlapped, H-6a), 1.808 (1H, dd, *J* = 14.8, 12.7 Hz, H-6b), 4.819 (1H, d, *J* = 3.6 Hz, H-7), 6.173 (1H, s, H-11), 1.471 (3H, s, H-18), 1.160 (3H, s, H-19), 7.313 (1H, overlapped, H-20), 6.475 (1H, d, *J* = 2.2 Hz, H-22), 3.014 (1H, dd, *J* = 12.8, 6.8 Hz, H-24a), 2.936 (1H, dd, *J* = 12.8, 7.9 Hz, H-24b), 2.612 (1H, m, H-25), 1.174 (3H, d, *J* = 7.0 Hz, H-27), 1.145 (3H, s, H-28), 1.110 (3H, s, H-29), 1.282 (1H, s, H-30), 6.788 (1H, d, *J* = 7.0 Hz, NH), 5.482 (1H, d, *J* = 7.0 Hz, H-2′ of PGME), 7.300 (3H, overlapped, phenyl protons of PGME), 7.139 (2H, overlapped, phenyl protons of PGME), 3.701 (3H, s, OCH_3_). HRMS(ESI) *m*/*z* [M + H]^+^ Calcd for C_39_H_44_O_8_N 654.30669, found 654.30615.

### 4.6. ^13^C NMR and ECD Calculation of ***1*** and ***2***

Conformation searches were performed at the MMFF94s force field. The conformers with population above 1% were optimized with density functional theory (DFT) at B3LYP/6-31G(d) level in gas and further optimized at the M06-2X-D3/Def2-SVP level of theory in Gaussian 16 program [39]. The conformers within 3 kcal/mol of global minimum were selected and calculated their ECD at *ω*B97XD/Def2-SVP level of theory, and NMR data at B97-2/pcSseg-1 level with IEFPCM model in chloroform. The shielding values of tetramethylsilane were calculated by the same methods (shielding values: C 193.9312, H 31.5135). ECD data were processed with SpecDis 1.71 [40] and plotted in Microsoft Office Excel 2019. NMR data were processed and plotted with Microsoft Office Excel 2019.

## Data Availability

Data are contained within the article and Appendix A.

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
