# Peer review of "Ganoaustralins A and B, Unusual Aromatic Triterpenes from the Mushroom Ganoderma australe"

_pharmaceuticals, 2022, doi:10.3390/ph15121520_

Round 1

Reviewer 1 Report

The manuscript entitled Ganoaustralins A and B, Unusual Aromatic Triterpenes from the Mushroom Ganoderma australe described the isolation, structural elucidation and bioactivity of two new triterpenes fearing an intriguing 6/6/6/5/6 skeleton from the Traditional Chinese Medicines. The chemical structures, including relative and absolute configurations, of A and B were well constructed with the efforts of NMR/MS spectroscopic analysis, chemical derivatization and NMR/ECD calculations. More interestingly, a plausible biosynthetic pathway of the characteristic aromatic ring in the two new compounds were also proposed. Finally, the reviewer thinks this manuscript to be a high quality work in the field of natural product and recommends its publication in the journal of Pharmaceuticals.

Author Response

The manuscript entitled Ganoaustralins A and B, Unusual Aromatic Triterpenes from the Mushroom Ganoderma australe described the isolation, structural elucidation and bioactivity of two new triterpenes fearing an intriguing 6/6/6/5/6 skeleton from the Traditional Chinese Medicines. The chemical structures, including relative and absolute configurations, of A and B were well constructed with the efforts of NMR/MS spectroscopic analysis, chemical derivatization and NMR/ECD calculations. More interestingly, a plausible biosynthetic pathway of the characteristic aromatic ring in the two new compounds were also proposed. Finally, the reviewer thinks this manuscript to be a high-quality work in the field of natural product and recommends its publication in the journal of Pharmaceuticals.

Re: Thank you very much for your positive comments.

Reviewer 2 Report

See attached file.

Author Response

General Comments. The authors have presented a modest paper on the isolation of two triterpenes, ganoaustralins A (1) and B (2), wherein the structure and determination of the absolute configuration of the stereoisomers as well as the PGME derivatization of 1 is discussed. Recommendations for manuscript improvement are found below.

Editorial Comments. Generally, the article is well written, but there are places where modifiers are omitted and sentence structure could be improved. A few examples are indicated below:

  1. Page 2, Introduction. Missing modifier. As reads, “…distributed in North Hemisphere [1,2].” Recommend changing to read, “…distributed in the North Hemisphere [1,2].”
  2. Page 2, Introduction. Sentence structure. As reads, “… australe, mainly dwells in tropic areas,…which can be distinguished…” Recommend changing to read, “…G. australe, which mainly dwells in tropic areas,…can be distinguished…”
  3. Page 2, Results. NMR clarification. As reads, “The 1D NMR spectra of 1…The 1D NMR spectra measured…” Recommend changing to read, “The 1D 1H NMR spectra of 1…The 1D 13C NMR spectra measured…”

Re: Thank you very much for your comments. The above-mentioned sentences were revised in the updated manuscript. The manuscript was been checked carefully. Other typos have also been revised.

Content comments.

  1. The characterization methods used by the authors provides excellent support for the identity, purity and absolute configuration of 1 and 2.

Re: Thank you very much for your positive comments.

  1. The use of computational chemistry (ECD) and the PGME derivatization method to assist in the determination of absolute configuration supports the authors’ structural and stereochemical assignments of 1.

Re: Thank you very much for your positive comments.

  1. Page 7, Scheme 1. The authors refer to the first step of the proposed biosynthetic pathway of 1 as an intramolecular aldol reaction. In fact, it looks to be an intramolecular Claisen condensation based on the presence of and subsequent loss of the methoxy group during the condensation. Recommend correcting the manuscript discussion regarding this step as well as the Scheme notation over the first arrow in the pathway.

Re: The Scheme I was revised by changing “aldol reaction” to “Claisen reaction”.

  1. It was noted that the 13C NMR data was not presented for the isomeric PGME-derivatives. While not essential for structural elucidation, recommend that the spectra be included as supplemental data.

Re: In this study, the yield of compounds 1 and 2 are very low. In order to save samples for more biological assays, the starting material used for PGME reaction usually be 0.5 mg. Although the yield of the PGME-products are high (> 90%), the quantities of the products are not enough to obtain the 13C NMR spectra by our facilities (We only have 600 MHz of NMR spectrometer, and is busying in testing samples. So, scanning such tiny amount of samples would take more than one week). On the other hand, for PGME method, we only need the 1H NMR spectra to find the differences between the chemical shifts of the protons around the chiral centers-to-be-determined. Moreover, the HRESIMS data confirmed the products are really PGME-derivatives.

Reviewer 3 Report

The authors are focusing on exploring Ganoderma fungus which is used in  Traditional Chinese Medicine. They successfully isolated and identified two new polycyclic triterpenes, ganoaustralins A and B. They confirmed the structure and the configuration of the two compounds using NMR and HRESIMS spectroscopic analysis. I have some general comments on their manuscript.

- Could you comment on the purity of the isolated compounds 1 and 2.

- Would you please comment on the solvent selected for the NMR analysis. For compound 1, you measured it first on chloroform, however, you noticed serious overlapped signals. Therefore, you repeated the measurement on pyridine that showed more clear signals. Have you also noticed the same for compound 2, could you comment on that.

- You mentioned that the biological activity of the two isolated compound were tested in a panel of assays, and compound 2 displayed moderate activity towards the b-secretase 1 (BACE1) enzyme. Firstly, I would recommend to add the tables of activities to the supporting information file, even if they are moderately active. The second point, have you thought about using any of the online servers for target predictions. It is quite important to know the biological target of the isolated compounds, so we can check their pharmacological action.

Author Response

The authors are focusing on exploring Ganoderma fungus which is used in Traditional Chinese Medicine. They successfully isolated and identified two new polycyclic triterpenes, ganoaustralins A and B. They confirmed the structure and the configuration of the two compounds using NMR and HRESIMS spectroscopic analysis. I have some general comments on their manuscript.

- Could you comment on the purity of the isolated compounds 1 and 2.

Re: The purity of compounds 1 is above 95%, while compound 2 is above 90%, by analysis of the 1H NMR spectra. These levels of purity are enough for spectral tests and biological assays.

- Would you please comment on the solvent selected for the NMR analysis. For compound 1, you measured it first on chloroform, however, you noticed serious overlapped signals. Therefore, you repeated the measurement on pyridine that showed more clear signals. Have you also noticed the same for compound 2, could you comment on that.

Re: The yield of compound 2 is low, changing the NMR solvent may cause sample loss during the handing procedures. Besides, the quality of 1H NMR spectrum of 2 is high, the splitting patterns of the peaks can be easily assigned. So we didn’t change the NMR solvent from chloroform to pyridine to measure other set of NMR spectra.

Generally speaking, the NMR spectra of a new skeleton usually measured in different NMR solvents to avoid misassignment. The “serious overlapped signals” in this study are refers to those signals that are critical for the structural assignment, especially which can support the presence of benzene ring, but partially overlapped with other signals. Secondly, the signals in 1H NMR spectrum are usually partially overlapped, not fully overlapped. The small chemical shift differences between the partially overlapped peaks are enough to differentiate the HMBC correlations. Thirdly, the quality of the spectrum is related to many other factors, such as the instrument parameters, the NMR solvents used, the shimming status of the spectrometer, the temperature…In this study, the quality of 1H NMR spectra of 1 is not as good as that of compound 2, though we have tried several NMR solvents, and spectrometer parameters for compound 1, the splitting of the peaks are rough. Finally, if we compare the 1H NMR spectra of these two compounds, they are not overlapped heavily. So to avoid misunderstanding, the sentence in Page 2 “The 1H NMR spectra of 1 recorded in chloroform-d presented serious overlapped signals, largely hampering the structural elucidation (Table 1).” was revised to “The 1H NMR spectra of 1 recorded in chloroform-d presented partially overlapped signals which were crucial for correct structural elucidation (Table 1)”

- You mentioned that the biological activity of the two isolated compound were tested in a panel of assays, and compound 2 displayed moderate activity towards the b-secretase 1 (BACE1) enzyme. Firstly, I would recommend to add the tables of activities to the supporting information file, even if they are moderately active. The second point, have you thought about using any of the online servers for target predictions. It is quite important to know the biological target of the isolated compounds, so we can check their pharmacological action.

Re: Thank you very much for the positive comments. The results of the biological activities for compounds 1 and 2 were added in the SI file. Because we don’t have the experience for target prediction by the online server, we didn’t use it this time. There is a biological activity screening center in Kunming Institute of Botany, Chinese Academy of Sciences, where provides a plenty of models for bioactivity screening. So, we usually send the samples there and choose different biological activities to test according to the literatures.
